# Simulation of the Landing Buffer of a Three-Legged Jumping Robot

**Yilin Yan** [1], **Katharine Smith** [1,*], **Alejandro Macario-Rojas** [1] and **Hongbo Zhang** [2,*]

1   Department of Mechanical, Aerospace and Civil Engineering, University of Manchester, Manchester M1 3BB, UK; yilinylouis@hotmail.com (Y.Y.); alejandro.macariorojas@manchester.ac.uk (A.M.-R.)
2   School of Mechanical and Power Engineering, East China University of Science and Technology, Shanghai 200237, China
*   Correspondence: kate.smith@manchester.ac.uk (K.S.); hbzhang@ecust.edu.cn (H.Z.)

**Abstract:** In recent years, the research of planetary exploration robots has become an active field. The jumping robot has become a hot spot in this field. This paper presents a work modelling and simulating a three-legged jumping robot, which has a powerful force, high leaping performance, and good flexibility. In particular, the jumping of the robot was simulated and the landing buffer of the robot was analyzed. Because this jumping robot lacks landing buffer, this paper verifies a method of absorbing landing kinetic energy to improve landing stability and storing it as the energy for the next jump in the simulation. Through the landing simulation, the factors affecting the landing energy absorption are identified. Moreover, the simulation experiment verifies that the application of the intermediate axis theorem helps to absorb more energy and adjust the landing attitude of the robot. The simulation results in this paper can be applied to the optimal design of robot prototypes and provide a theoretical basis for subsequent research.

**Keywords:** planetary exploration; jumping robot; bionic landing buffer; intermediate axis theorem; modelling and simulation

## 1. Introduction

### 1.1. Motivation

There is a growing interest in space exploration. Robotic technology is a disruptive technology for space exploration, which has received wide attention [1]. A robot that can jump, called jumping robot in this paper, is advantageous over other locomotion robots, such as wheeled and legged robotic system, in terms of its ability to "walk" over high bumped terrains, on planets such as Mars and the Moon [2]. Besides applications in space exploration, jumping robots also have applications in freight transportation and military battlefields [3].

The space systems research group at the University of Manchester has carried out several activities around robotic planet exploration, including a three-legged jumping robot. It could complete all the typical action processes of intermittent jumping and showed high energy conversion efficiency and jumping ability. There are still some defects that limit its performance, including unstable landing buffer and discontinuous jumping motion.

### 1.2. Related Works—Landing Energy Absorption

This three-legged robot's jumping mechanism was inspired by the frog and improved by adding one more leg to make it have greater jumping ability. Due to the lack of landing buffer, its hard landing made it impossible to land stably, so it has to be reset in order to prepare for the next jump. This paper is inspired by the frog's landing, as the anticipatory hindlimb flexion of the frog during the aerial phase is a critical feature for mechanically stable landing [4], so this robot may also flex its legs during the aerial phase to make

it have a stable landing buffer. The tendons of many animals not only store energy of locomotion in the form of potential energy when standing, but also play an important role at touchdown [5]. The elastic tendons in legs can deform to absorb the landing impulse shock, and the energy stored in deformed tendons can be reused for the next jump. The tendons work as power amplifiers and achieve metabolic energy conservation. Chen et al. studied locusts' land buffering mechanism; Romano et al. studied the relationship between locusts' legs postures and body displacement through kinematic analysis, and proposed that multiple elastic legs can reduce the maximum contact force, increase the energy absorption per distance, and achieve better landing buffer [6–8]. For the single-legged continuous jumping robot KenKen [9], its leg acts as both jumping and landing leg, which can absorb part of the landing shock and store energy. The elastic spring placed in its legs works similarly to tendons in real animals' landing. By changing the leg posture in the air, the leg spring can absorb a large impulse at touchdown and store energy for the next jump. This method of changing leg posture in the air can be applied to improve the landing buffer of the three-legged jumping robot, while also storing energy for the next jump.

*1.3. Related Works—Intermediate Axis Theorem*

Due to lack of aerial righting, the jumping robot tilted in the air after taking off, which was another factor that affected its stable landing. The mini robot Grillo lacked aerial righting mechanisms and would pitch down in the air while jumping forward, which eventually made it unable to land stably. By adjusting the take-off angle and strength, the researchers increased the robot's tendency to pitch down and made it somersault forward, making it flip to a stable landing attitude [10]. In the literature, a method of controlling the reversal of a rotating spacecraft is proposed [11]. This method is facilitated by the Dzhanibekov effect or the intermediate axis theorem (IAT) [12,13]. This method allows for the activation of the flipping motion of the spacecraft, which is initially experiencing its stable rotation. In the same way, this method can make the rotating robot flip in the air, and make it flip to a vertical attitude to land, which solves the problem of tilted landing. At the same time, the impact of applying IAT on the landing energy absorption will also be researched. A multi-DOF jumping robot can perform normal jumps and somersaults. The landing ground reaction force of its somersault is 1.7-times that of a normal jump [14]. Therefore, after the robot flips in the air, it will receive a larger ground reaction force when it lands, so that more landing energy can be absorbed.

*1.4. Related Works—Modeling and Simulation*

In this paper, the robotic simulation system will be used for research instead of physical experiments. The mechanical model of the somersault jumping robot mentioned above was developed based on the design parameters determined by the simulation experiments [14]. A physical simulation model developed by its research and development team is referred to and modified in this paper to simulate the jumping kinematics and analyze the landing buffer of the jumping robot. The parameters obtained by the simulation will have a strong guiding role for the subsequent optimization of the robot in physical experiments [15].

*1.5. Contribution*

Herein, we present a modified simulation model of the three-legged jumping robot for a landing simulation, verifying two methods to improve landing buffer:

(1) The first method is to change robot's leg posture in the air by compressing the robot and bending its leg, so that the leg springs can absorb the impulse at touch-down and transfer its kinetic energy into potential energy for the next jump. This method is verified by simulation experiments and the landing leg's posture that can absorb the most landing energy at different drop heights is obtained.

(2) The second method is to add the application of IAT on the basis of the previous method. In this method, the robot also bends its legs in the air to absorb the landing energy, but after it takes off, a constant rotation speed is added to make it flip in the

air, which is the phenomenon described by IAT. By finding a suitable rotation speed, it can finally flip to a vertical attitude to land, so that the existing tilting landing attitude can be corrected. The simulation experiment verifies this method and obtains the most suitable rotation speed under different falling heights, as well as the influence of this method on landing energy absorption compared with the first method.

The research objectives for this paper are as follows:

- Develop a simulation model that meets the main characteristics of the jumping robot and the requirements of the above two methods;
- Through simulating the landing dynamics of the jumping robot, identify the factors that affect landing energy absorption and explore the application of the IAT.

The remainder of the paper is organized as follows. Section 2 introduces the three-legged jumping robot, Section 3 gives the simulation model prototype, Sections 4 and 5, respectively, present the simulation experiments of the two methods and the discussion of their results and, finally, Section 6 offers the conclusions.

## 2. Three-Legged Jumping Robot

The main research objective is a three-legged jumping robot developed by the MACE (Department of Mechanical, Aerospace and Civil Engineering) space robotics team at the University of Manchester. The bioinspired articular structure design of the jumping robot was commonly observed from the minimalistic hopping robot by Hale as an initial reference. By imitating the jumping process of the frog, a design combined with the 6-bar geared mechanism with springs was developed. The robot completes the energy accumulation by stretching the spring after being compressed. The compression of the robot is achieved by rotating the power screw through the motor and gearbox on the top to shorten the leg compression length. After the energy accumulation is complete, the spring is released to complete the jump. The robot can jump up to a height of 80 cm and a horizontal distance of 1.8 m. Its mass is about 1.3 kg; the heavy weight limits its jumping performance [16].

By imitating the prototype developed by Hale, the team invented the 3-legged jumping robot (see Figure 1). The jumping robot weighs 75.5 g and is 23 cm high, the jumping height stabilizes at 41 cm and the highest point the robot can reach is about 66.5 cm [17]. Based on the mechanism of the minimalistic hopping robot by Hale, the team added a third leg and two more springs for better jumping ability. To avoid the prototype being too heavy, metal springs were replaced with rubber straps, carbon fiber strips are used for all legs, and all custom part geometries are fabricated using PLA (PolyLactic Acid) filament material and FDM (Fused Deposition Modelling) printing. These advanced materials and additive manufacturing technology makes the robot lightweight and robust enough. In addition, a third leg makes a more robust structure and makes the future replacement of stiffer spring materials possible for more jump energy accumulation.

As shown in Figure 2, the three legs of the robot connect the upper and lower body. By imitating the design of Hale's robot, on the upper body, the motor is at the very top, and the gearbox that amplifies the torque of the motor is fixed under it. Unlike Hale's robot, the upper and lower bodies of this robot are connected by a few thin wires instead of a power screw. One end of the ropes is wrapped around a winch below the gear box, and the other end is fastened to the upper part of the lower body. The stick under the winch is used to trigger the mechanism that separates the upper half of the lower body. The upper part that fixed ropes is locked on the lower body during compression and is separated from the lower body after the separation mechanism is triggered. All the components of the electronics are fixed around the upper body, which include an Arduino Bluetooth 4.0 Board, a 9-Degrees-Of-Freedom absolute orientation inertial measurement unit, a lithium polymer battery, and power management electronics.

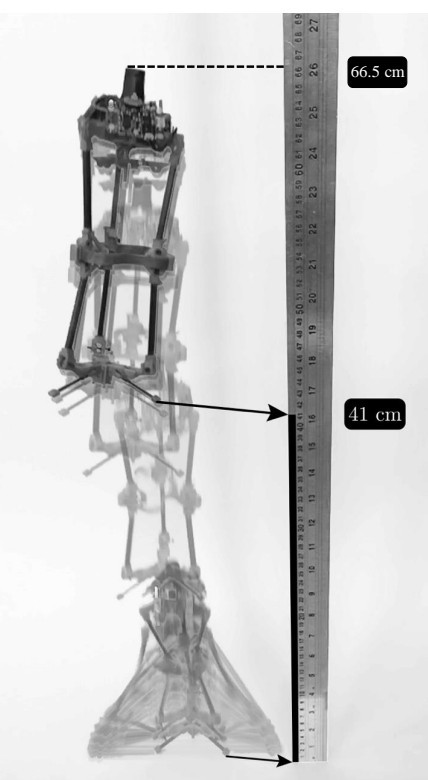

**Figure 1.** The three-legged jumping robot and its vertical jumping recording [17].

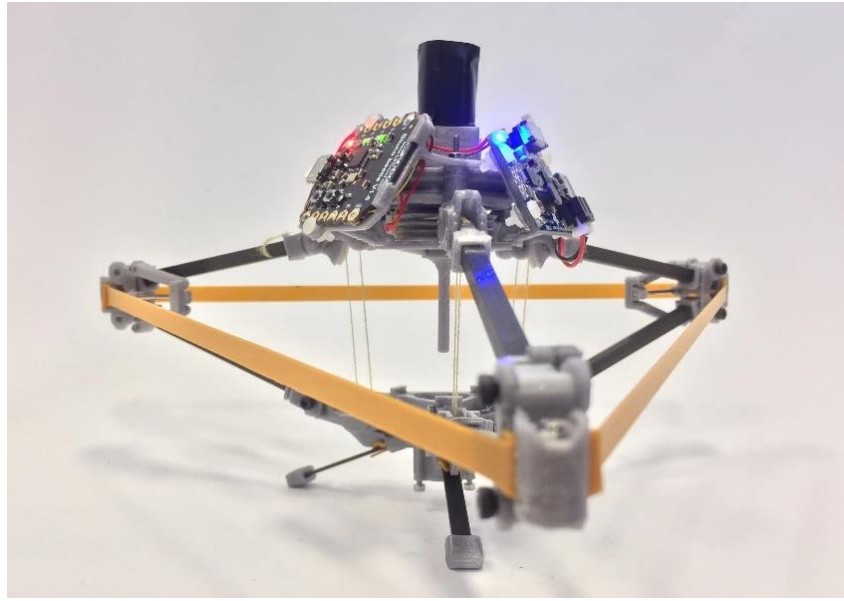

**Figure 2.** The three-legged jumping robot in compressed position [17].

The jumping process of the robot consists of three parts, which are energy accumulation process, take-off process and landing process. As shown in Figure 3, the energy accumulation process is from phase (a) to phase (b), and the robot is compressed by rotating the winch to shorten the length of the ropes between the upper and lower body. During the compression process, the yellow straps are stretched by three joints to accumulate jumping energy. The take-off process is shown from phase (b) to phase (c). After the separation mechanism is triggered, the upper parts that fixed the ropes are separated from the lower body. Under the decompression of the yellow strip, the robot starts to take off. Under

the action of gravity, the robot falls after reaching the highest point, which is the landing process shown from phase (c) to phase (d). As shown in phase (d), the robot was landing on one of the legs, which means that the landing attitude of the robot was inclined. Due to a lack of aerial righting, the robot deviated from the vertical axis when it was in the air, and the angle of deviation reached the maximum when it landed.

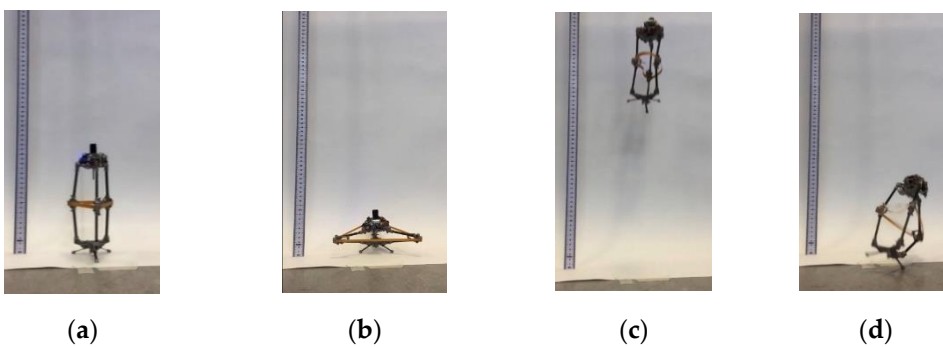

| (a) | (b) | (c) | (d) |

**Figure 3.** Jumping process of the prototype: (**a**) uncompressed state; (**b**) compressed state; (**c**) take-off state; (**d**) landing state.

## 3. Referred Simulation Model

The jumping robot model for landing simulation will be referred from the MACE space robotics team. The simulation model is the analog of the simulated object or its structural form and it can be a physical model or a mathematical model [18]. In this paper, the referred simulation model prototype is a physical model. The researchers generated a simulation model of the jumping robot through MATLAB Simulink, based on the physical characteristics of the jumping robot, and the block diagram is shown in Figure 4. The jumping robot simulation is an approximate imitation of the operation of the robot.

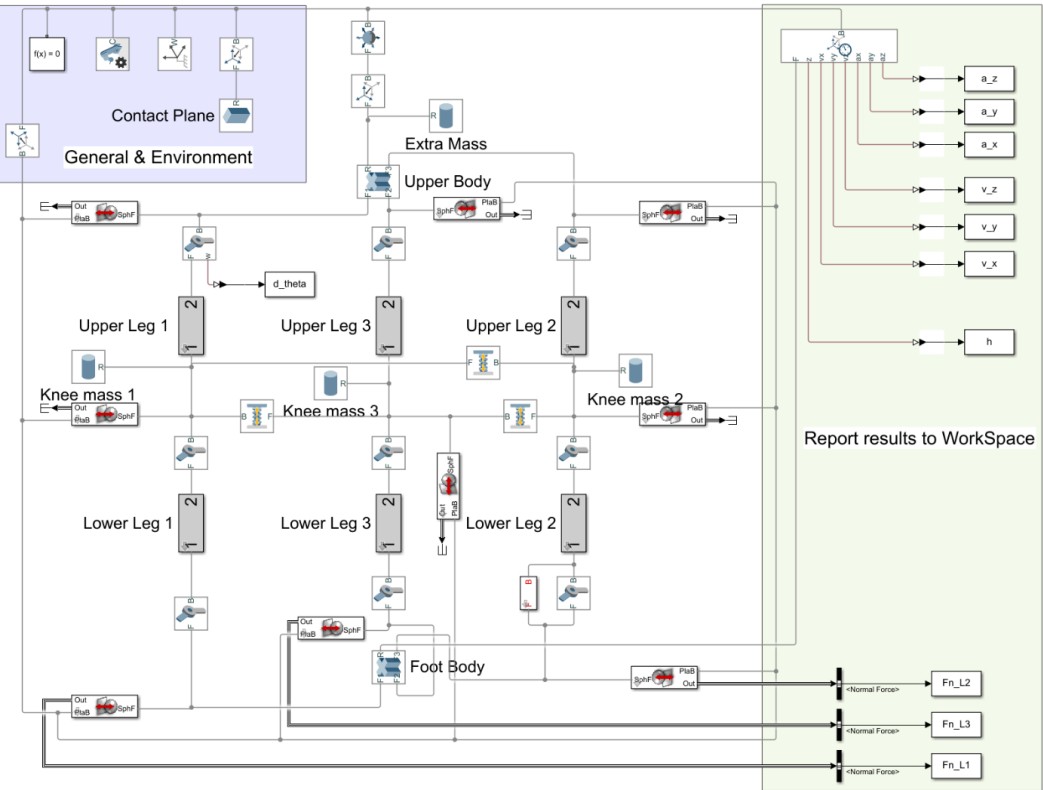

**Figure 4.** Referred Simulink block diagram.

The simulation model prototype will be divided into four parts for detailed introduction; they are the jumping rover model, parameter input, output channels, and data analysis.

As shown in Figure 5, the simulation model retains the main features of the jumping robot and simplifies the parts that are irrelevant to the robot's dynamics. Both the simulation model and the jumping robot consisted of three legs, communicating an upper and a lower body section. The elastic strap attached to three middle-leg hinges was replaced by three invisible springs with the same stiffness in the simulation model. Each hinge was simplified to an equivalent point mass in the knee joint. The upper and lower bodies were simplified into two triangular plats of the same dimension. The mechanical power system for the upper body was simplified as a cylinder with the same mass and similar dimensions. The three landing legs under the lower body were ignored. To simplify the jumping rover model while still establishing a fairly accurate model as much as possible, several assumptions have been made. The physical structure of the jumping rover is a rigid frame, so that its elastic vibration and deformation are not considered. The aerodynamic effects, such as air resistance, were not considered. The internal friction of the robot is ignored, such as the friction between the joints. Since the energy accumulation process does not participate in the simulation of jumping and landing, the simulation prototype omits it, in order to simplify the simulation model, so the simulation of the robot starts from the compressed state, ready to take off. To define the compressed state of the robot at the start of the simulation, the acute angle formed by the upper or lower leg and the vertical is defined as the squat angle, which is the θ in Figure 5.

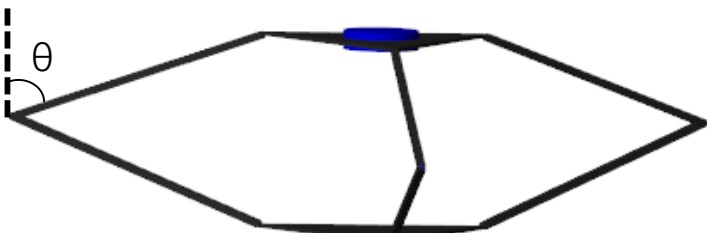

**Figure 5.** Schematic of the simulation robot in compressed state.

Some physical constants, model parameters and initial conditions were treated as input and integrated in MATLAB Editor, which was convenient for editing these inputs directly. Meanwhile, some parameters that were not frequently changed are directly defined in the corresponding modules in the block diagram, such as the condition of the contact plane.

As shown in Figure 4, the output channels of the model are distributed in the green area on the right, and the data of the robot's velocity, acceleration and height and the forces on the three legs are reported to MATLAB WorkSpace.

After the simulation results are reported to the WorkSpace by the output channels, these data can be analyzed in different ways. Figure 6 shows the resulting curves of height reached by the robot versus time, plotted from the data output from the simulation prototype. As can be seen from the comparison photos at the beginning, at the start of the simulation, the robot was already compressed and ready to take off. Before the simulation was started, the robot was input with a squat angle of 80 degrees, which was close to the experimental squat angle in the compressed state. In this way, the robot in the simulation could be considered to obtain the same energy accumulation as the experimental robot. Another comparison photo was taken when the robot reached the highest point. It can be seen that the highest point reached by the robot in the simulation was 0.6968 m and the highest point in the experiment was 0.665 m, so the deviation rate was 4.78%. The deviation rate is acceptable, so the simulation model prototype proved to be credible.

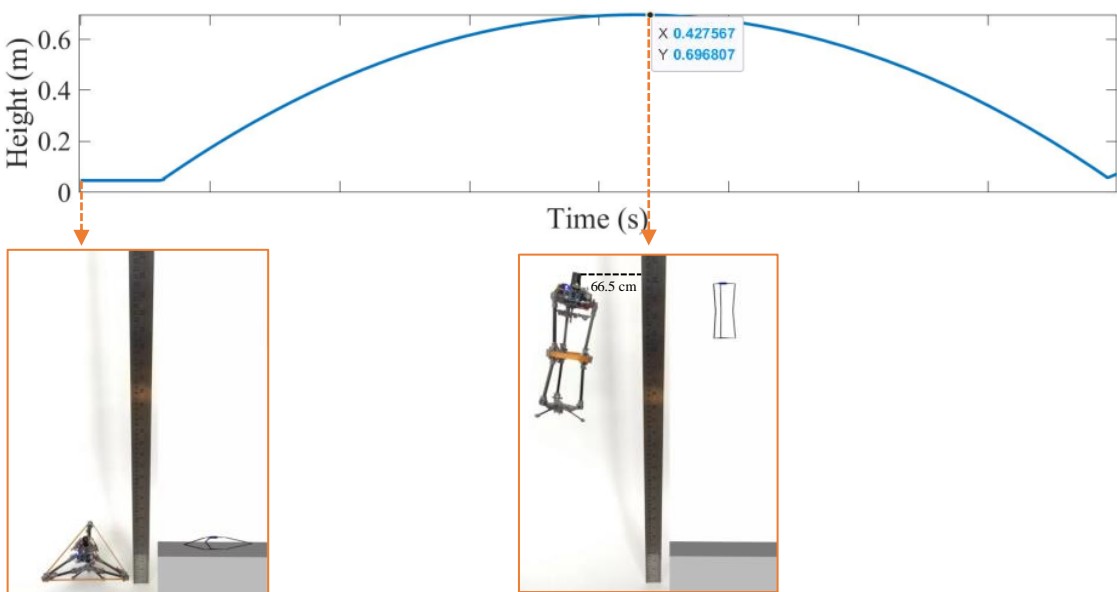

**Figure 6.** The height reached by the robot in the simulation and the corresponding two sets of comparison pictures.

## 4. Landing Energy Absorption Simulation

The basic idea of landing energy absorption is to change the robot's leg posture in the air by compressing the robot and bending its leg, so that the leg spring can absorb the impulse at touch-down and transfer its kinetic energy to potential energy for the next jump. The robot's squat angle can be used to define the compressed legs posture in the air. The larger the squat angle, the more the robot is compressed. Since this robot is designed to explore planets, the simulation experiment in this paper will be carried out under lunar gravity. Therefore, this chapter aims to find out the relationship between the model's landing squat angle and the energy absorption under lunar gravity.

### 4.1. Methodology

Since the speed of the robot is zero when it reaches the highest point, the landing process can be regarded as a releasing process, where the model is dropped from the highest point. The landing legs' posture of the jumping robot is represented by the landing squat angle. At the beginning of the simulation, the robot was compressed to the input landing squat angle and was placed at the designated height to wait for the fall. To prevent it from decompressing during the fall, the squat angle of the robot in the simulation was constrained so that it couldn't exceed the input squat angle. Experimentally, the maximum compressed state of the robot is 80 degrees squat angle. Therefore, the squat angle during falling was limited from the input landing squat angle to 80 degrees.

The research object of this experiment is the energy absorbed after landing, which could be represented by the elastic potential energy difference of the springs. Since the simulation model prototype cannot directly output the elastic potential energy of the spring, the elastic potential energy needs to be converted by the force on the spring [19], which is derived as follows:

$$U = \frac{1}{2}k\Delta x^2 = \frac{1}{2}k\left(-\frac{F_x}{k}\right)^2 = -\frac{F_x{}^2}{2k} \tag{1}$$

where $U$ is the elastic potential energy, $k$ is the lift-dependent drag coefficient, $\Delta x$ is the deformation of the spring, $F_x$ is the elastic force.

When the falling height is constant, the relationship between the landing squat angle and energy absorption can be obtained by recording the elastic potential energy changes of different landing squat angles.

To get a broader sense of the relationship between energy absorption and landing squat angle, the relationships under different dropping heights need to be obtained. Hence, 15 sets of simulations at different release heights were conducted. The 15 drop heights were taken from the jump heights of the corresponding 15 launching squat angles. These 15 angles started from 10 degrees, and then each angle was increased by 5 degrees until 80 degrees, so, 10 degrees, 15 degrees . . . 80 degrees. Therefore, the first step of this simulation was to determine the 15 corresponding jump heights under lunar gravity. After the jump height was obtained, the drop height of 15 sets of simulations on energy absorption could be determined. At each drop height, different landing squat angles were input to obtain the corresponding elastic potential energy difference. In the results for each drop height, the drop squat angle that absorbed the most energy was the best landing squat angle of a certain height.

### 4.2. Simulation Procedure

(1) To get the jumping height of the model under the selected 15 launching squat angles under lunar gravity, in the input parameters, the gravity was changed to lunar gravity, which was considered to be 0.166-times Earth's gravity. Next, the compression angle of the model was set as the 15 selected angles, respectively. By running the simulation 15 times, the jump heights of each launching squat angle were obtained.

(2) The simulation model for landing energy absorption, shown in Figure 7, was observed from the prototype of the simulation model and was modified according to the above methodology. Since the data output in the prototype was useless in this simulation, the output channels in the prototype were replaced by three channels that output three spring forces. The Sense Force function in the three Springs and Damper Force block was enabled, so that the elastic force on each spring could be output.

(3) To prevent the robot from decompressing during the fall, the landing squat angle was limited from a certain input landing squat angle to 80 degrees, though the Rotational Hard Stop Friction block marked in Figure 7.

(4) After the data matrix of the elastic force of the three springs over time was output to the WorkSpace, they were added together and (1) were applied to obtain the data matrix of total elastic potential energy changed with time, and then, by calculating the difference of the elastic potential energy before and after landing, so that the energy absorbed during landing could be obtained. The above calculation process was implemented by the codes in MATLAB, which can be found in the 'Energy absorption' paragraph in the Appendix A.

(5) The initial squat angle and height of the robot could be set in the MATLAB codes of the prototype, which were used to set the landing squat angle and height of the robot. At each drop height, different landing squat angles were input to obtain the corresponding elastic potential energy difference. The landing squat angle was set from 0 degrees to 80 degrees, accurate to 1 degree. Therefore, at each height, the elastic potential energy differences corresponding to the 81 landing squat angles were generated. In this way, the previously obtained 15 drop heights were sequentially substituted into the codes, so that the relationship between the landing squat angle and energy absorption at different heights was obtained. Finally, the resulting curves for the 15 sets of data were plotted in one graph.

### 4.3. Results and Analysis

#### 4.3.1. Jumping Height

Table 1 gives the simulation jump height under different launching squat angles, which provide data for the following simulations.

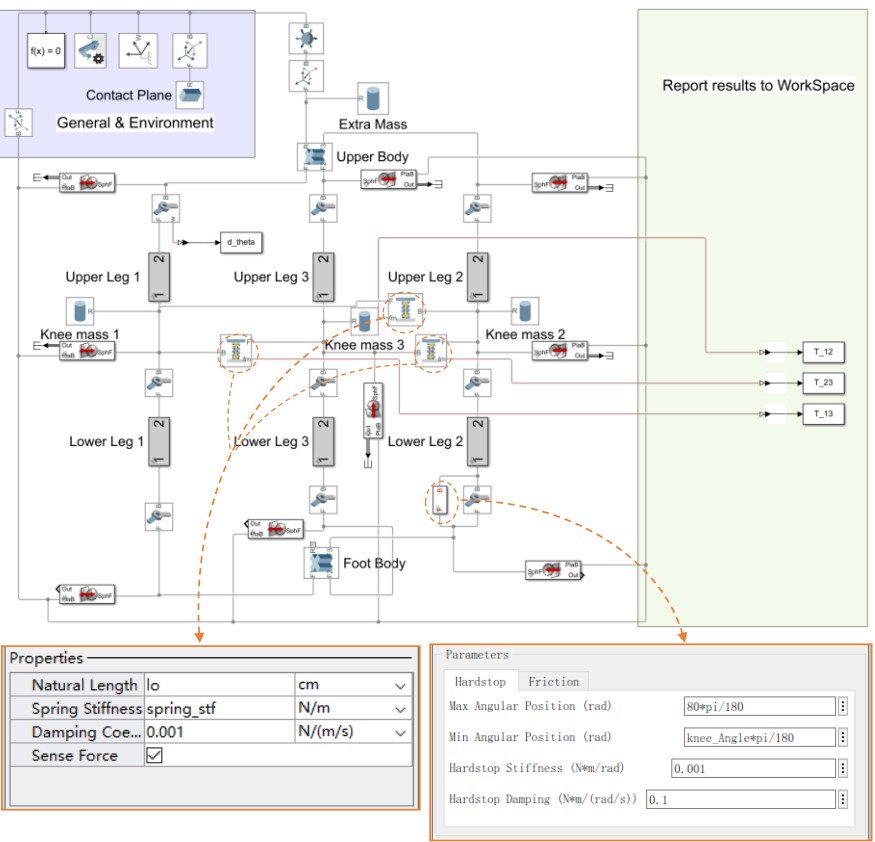

**Figure 7.** Modified Simulink block diagram for landing energy absorption.

**Table 1.** Simulated jump height at different launching squat angles.

| Launching Squat Angle (Degree) | Jumping Height (m) |
| --- | --- |
| 10 | 0.0956 |
| 15 | 0.1588 |
| 20 | 0.3655 |
| 25 | 0.5986 |
| 30 | 0.9207 |
| 35 | 1.3109 |
| 40 | 1.6858 |
| 45 | 2.0853 |
| 50 | 2.5177 |
| 55 | 2.9327 |
| 60 | 3.3263 |
| 65 | 3.6641 |
| 70 | 3.9694 |
| 75 | 4.2078 |
| 80 | 4.3718 |

4.3.2. Landing Energy Absorption

Figure 8 shows the resulting curves of the elastic potential energy absorbed at different landing squat angles and there are 15 curves, representing 15 drop heights (the legend describes the corresponding launching squat angle and drop height of each curve). Taking the 80 degrees launching squat angle as an example, the values at the beginning of the curve are very small and close to 0, then the curve increases first and decreases after reaching the peak. In the process of increasing, the slope of the curve first becomes smaller and then becomes larger, and the slope is almost unchanged in the process of decreasing. The other

curves share the same characteristics, and when the drop height is higher, the best landing squat angle is smaller, and the maximum landing energy absorption is greater.

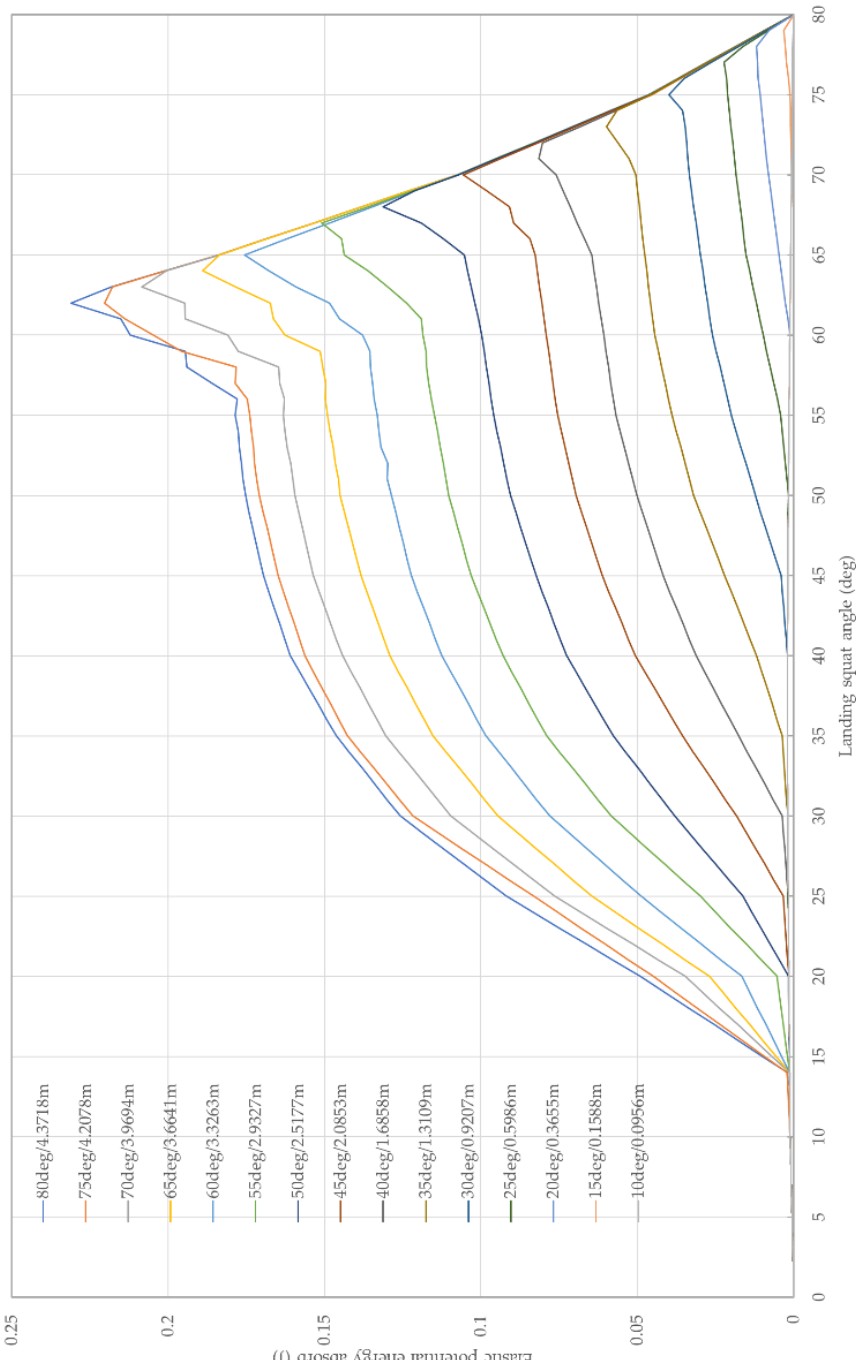

**Figure 8.** The elastic potential energy absorbed at different landing squat angles from different dropping heights.

Since the 80-degree launching squat angle is the closest to the experimental working condition of the jumping robot, the results obtained on this curve are analyzed. The obtained results analysis is shown in Table 2. When the landing squat angle is 62 degrees, the most landing energy is absorbed, which is 0.2310 J. The absorbed energy saves 13.36% energy for the next jump.

**Table 2.** Results analysis of the dropping height at 4.3718 m.

| Parameter | Unit | Data |
|---|---|---|
| Compression angle | deg | 80 |
| Compression energy | J | 1.7294 |
| Jumping height | m | 4.3718 |
| Best landing squat angle | deg | 62 |
| Most energy absorbed | J | 0.2310 |
| Energy saving percentage | % | 13.36 |

## 5. Landing Simulation with the Application of the Intermediate Shaft Theorem

Since the three-legged jumping robot has no aerial righting, the landing attitude is inclined. After landing, it needs to be reset manually to conduct the next jump. The application of IAT is to make the attitude of the robot vertical when landing, so that the previously verified landing buffer methodology can be applied. By adding an initial rotation velocity to the jumping robot at the beginning of the simulation, we can identify the conditions for the occurrence of the phenomenon of IAT and investigate how the application of IAT is helpful for landing energy absorption and landing attitude control. Due to the phenomenon of unstable rotation flipping, and to control the robot's landing attitude, the goal of the experiment was set, under the condition that the landing squat angle remains unchanged, by changing the initial angular velocity to make the jumping rover flip twice in the air and then land vertically with its the lower body.

### 5.1. Methodology

The IAT is the result of the overall mass distribution of the robot under the action of the initial angular velocity, which leads to unstable rotation around the intermediate axis. Therefore, to increase this instability and make this phenomenon more likely to occur, the cylindrical counterweight on the upper body of the robot was replaced by a cuboid, as shown in Figure 9. According to the IAT, in the state of unstable rotation of the object, the cycle time of each flip is constant when the rotation speed is constant. Therefore, if the robot flips twice in the air after take-off and then lands vertically with its lower body, this means that during the ascending and descending process, the robot flips once, respectively. Therefore, this simulation was simplified to drop the robot from a given height in an upside-down attitude.

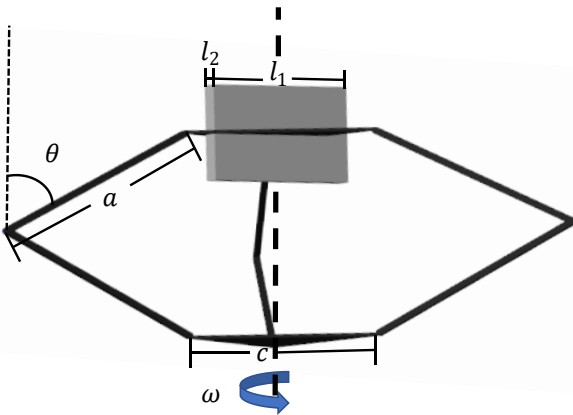

**Figure 9.** Schematics of the modified model for the IAT.

Then, we changed the initial angular velocity to make it flip once in the fall and land vertically with its the lower body. To show the landing attitude of the model, the attitude of the model was expressed by the unit vector formed from the midpoint of the lower body

to the midpoint of the upper body. According to the unit vector formula [20], the robot's unit vector can be conducted by the following Equation:

$$\hat{v} = \frac{\vec{v}}{\left|\vec{v}\right|} = \frac{(x, y, z)}{\sqrt{x^2 + y^2 + z^2}} \tag{2}$$

where $\hat{v}$ is the unit vector, $\vec{v}$ is the vector, $\left|\vec{v}\right|$ is the magnitude of the vector, and $x, y, z$ are the coordinates of the vector. The motion of the model could be regarded as a rigid motion with six degrees of freedom, which included three types of rotation around three axes and three linear motions along three axes. By plotting a three-dimensional figure of the orientation of the attitude vector on the three axes, it could express the trajectory of the vector in the fall [21].

Next, by plotting the three direction cosines of the unit vector with time, it could show the change of the vector around three axes during the fall [22]. The three direction cosines can be conducted using the following Equations:

$$\begin{aligned}
\alpha = \cos a = \frac{x}{\left|\vec{v}\right|} = \frac{x}{\sqrt{x^2+y^2+z^2}} \\
\beta = \cos b = \frac{y}{\left|\vec{v}\right|} = \frac{y}{\sqrt{x^2+y^2+z^2}} \\
\gamma = \cos c = \frac{z}{\left|\vec{v}\right|} = \frac{z}{\sqrt{x^2+y^2+z^2}}
\end{aligned} \tag{3}$$

where $\alpha$, $\beta$, $\gamma$ are the direction cosines for the X axis, Y axis, Z axis, and $a, b, c$ are the angles of the vector with respect to the three axes. Through these two plots, the flip of the model during the fall could be explained, and the landing postures of the model were identified.

Before being released, the model was compressed to the best landing squat angle that was obtained in the previous simulation, so that it could absorb the most energy when landing. As in the previous simulation, the squat angle of the robot was restricted to prevent it from decompressing during the fall.

In the simulation, an initial rotation speed was added to the robot. The rotation speed that could make the robot flip once in the fall and land with a vertical attitude was what needed to be obtained in this simulation experiment. Since the attitude vector is a unit vector, if the robot lands vertically, the direction cosine is 1 on the X and Y axes, and 0 on the Z axis. This initial rotation speed was tried from the input of 0 rad/s and the landing direction cosines of the robot were observed. We tried increasing the rotational speed and running the simulation repeatedly until the required speed was tried out.

After releasing, the data of the energy absorbed during the landing was collected, so that the change in the landing energy absorption after applying IAT could be obtained. It is worth noting that after the initial angular velocity was applied, the energy input for take-off had increased. Without considering the energy loss, the input energy that made the model rotate equaled its rotational kinetic energy. Therefore, the rotational kinetic energy of the model at the certain initial angular velocity needed to be identified. As shown in Figure 9, the basic parameters of this structure are the leg segments a, the leg–body articulation distance c, the squat angle $\theta$, the cuboid's length $l_1$, the cuboid's width $l_2$ and the initial angular velocity $\omega$. The rotational axis and direction are displayed. The derivation of the total moment of inertia is as follows.

For the upper block, since the axis of rotation is the central axis of it, its moment of inertia can be expressed by the following Equation [23]:

$$I_1 = \frac{1}{12} m_1 \left( l_1{}^2 + l_2{}^2 \right) \tag{4}$$

where $I_1$ and $m_1$ are the moment of inertia and mass of the upper block, respectively.

For the cuboid, since the axis of rotation is the central axis perpendicular to the triangle plane, its moment of inertia can be expressed by the following Equation:

$$I_2 = \frac{1}{12} m_2 c^2 \tag{5}$$

where $I_2$ and $m_2$ are the moment of inertia and mass of the triangle plate, respectively.

For one leg segment, which can be treated as a thin rod, since its rotation axis does not pass through its center, the parallel axis theorem needs to be applied. First, when the rotation axis passes through its center, the angle between the rotation axis and the direction perpendicular to the leg can be expressed as the landing squat angle, so that its moment of inertia can be expressed as the following Equation:

$$I_{3c} = \frac{1}{12} m_3 a^2 \sin^2 \theta \tag{6}$$

where $I_{3c}$ and $m_3$ are the moment of inertia and mass of one leg segment, respectively, with its rotation axis passing through the center of mass.

Then, according to the parallel axis theory [24], when the rotation axis is translated to the center of the model, its moment of inertia can be expressed by the following Equation:

$$I_3 = I_{3c} + m_3 L_{PAT}{}^2 = \frac{1}{12} m_3 a^2 \sin^2 \theta + m_3 \left( \frac{1}{2} a \sin\theta + \frac{\sqrt{3}}{3} c \right)^2 \tag{7}$$

where $I_3$ is the moment of inertia of one leg segment applied parallel axis theorem, and $L_{PAT}$ is the distance of parallel movement.

For a point mass on the knee, its moment of inertia can be expressed by the following Equation [25]:

$$I_4 = m_4 r_4{}^2 = m_4 \left( a \sin\theta + \frac{\sqrt{3}}{3} c \right)^2 \tag{8}$$

where $I_4$ and $m_4$ are the moment of inertia and mass of one knee point, respectively.

Since there is one cuboid, two triangular plates, six leg segments, and three knee point masses, the total moment of inertia of the model can be expressed by the following Equation:

$$I_{model} = I_1 + 2I_2 + 6I_3 + 3I_4 \tag{9}$$

where $I_{model}$ is the total moment of inertia of the model.

Therefore, the rotational kinetic energy of the model can be expressed by the following Equation [26]:

$$E = \frac{1}{2} I_{model} \omega^2 \tag{10}$$

where $E$ is the rotational kinetic energy of the model.

To obtain the relationship between the falling height and the target angular velocity, the model was dropped at the 15 heights in the previous simulations, and the initial angular velocity at which the robot landed in a vertical attitude of each height was obtained.

### 5.2. Simulation Procedure

The simulation model considering the IAT is shown in Figure 10, which was observed from the previous simulation model for landing energy absorption and was modified according to the above methodology. The detailed modifications are illustrated in the following simulation process.

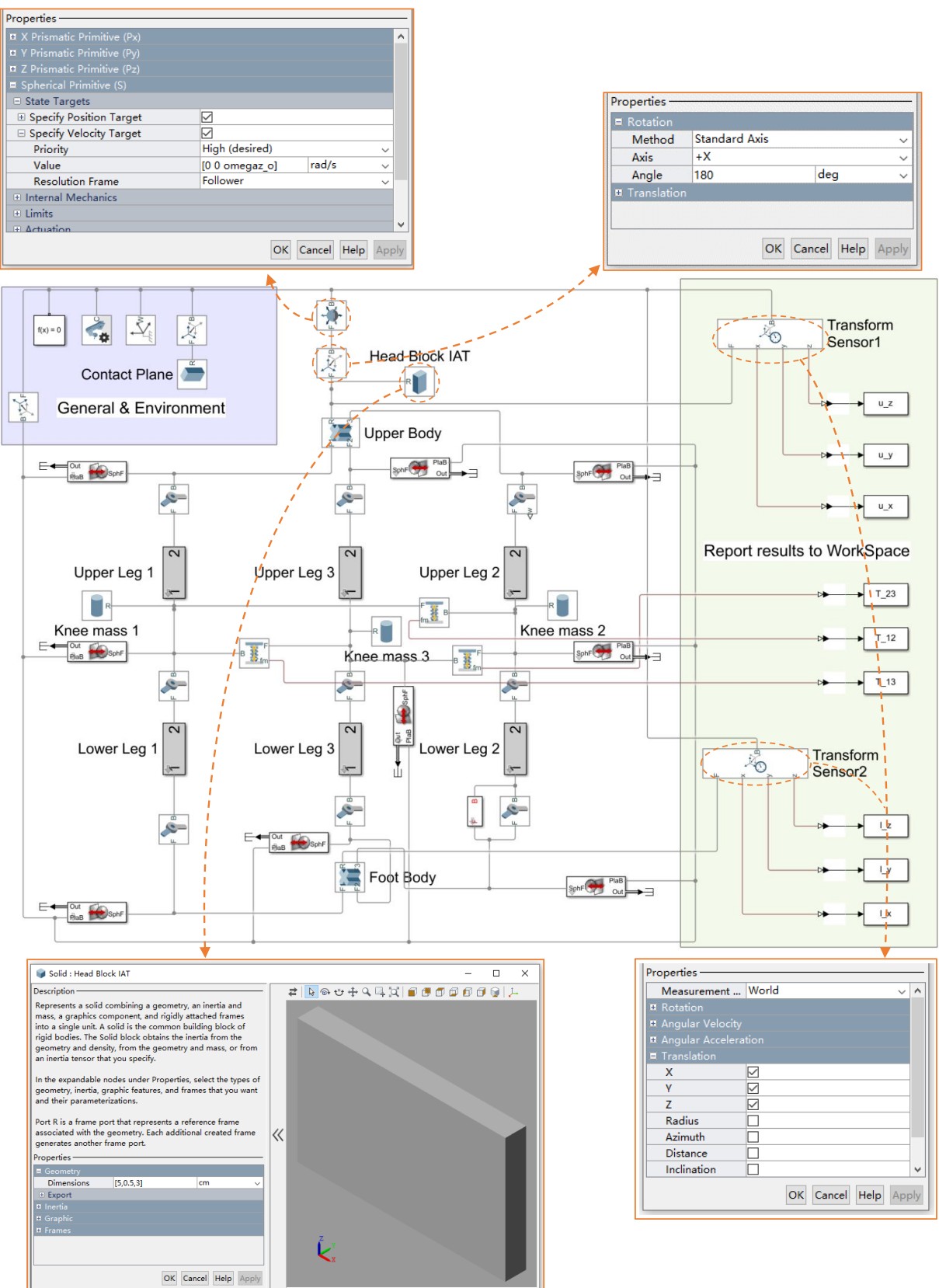

**Figure 10.** Modified Simulink block diagram for the IAT simulation and four marked modules.

(1)     First, the cylinder used to simplify the mechanical power system in the upper body of the model was replaced with a cuboid. The dimensions of the cuboid were defined as 5 cm × 0.5 cm × 3 cm by the Solid block marked out in Figure 10.

(2)     Then, the model was turned upside down at the initial position. Through the Rigid Transform block marked in Figure 10, the model was rotated 180 degrees around the X axis.

(3)     Next, the initial angular velocity of the model was added in the 6-DOF Joint block marked in Figure 10. The model was added with an angular velocity 'omega_o' around the Z axis. By changing the magnitude of 'omega_o' in the MATLAB Editor, the initial angular velocity was controlled as an input parameter.

(4)     As shown in Figure 10, the coordinates of the two midpoints of the upper and lower bodies relative to the ground were output through two Transform sensor blocks. After outputting the data matrix of the coordinates of the vectors over time to the WorkSpace, by applying (2), the unit vectors over time were calculated and a three-dimensional figure of the trajectory of the unit vector was plotted. By applying (3), the three directional cosines over time were calculated and plotted together on a graph. The calculation process and plotting instructions were implemented by the MATLAB codes, which can be found in the 'Spiral path' and 'Direction cosine' paragraphs in the Appendix A.

(5)     The best landing squat angles and their corresponding drop heights, obtained in the previous simulation experiments, were input into the simulation through MATLAB codes. When the landing squat angle and drop height remain unchanged, we experimented to input different rotation speeds until the direction cosines were output 1 for $\alpha$ and $\beta$, and 0 for $\gamma$. The initial rotation speed was tried from 0 rad/s, increasing by 0.1 rad/s each time and running the simulation repeatedly. After each simulation, the landing direction cosine of the robot was observed, until the direction cosines were output 1 for $\alpha$ and $\beta$, and 0 for $\gamma$. When the condition of the direction cosine was satisfied, the corresponding vector direction trajectory graph was checked to prove that the model had only flipped once, so that the input angular velocity was proved to be desired.

(6)     After the required angular velocity was obtained, the input energy required to achieve the angular velocity was calculated according from (4) to (10). The codes for calculating rotational kinetic energy can be found in the 'Rotational kinetic energy' paragraph in the Appendix A.

(7)     According to the above steps, the desired initial angular velocities at 15 different drop heights and the corresponding rotational kinetic energy of the robot were determined.

*5.3. Results and Analysis*

Table 3 shows the desired angular velocity at different drop heights. As shown in the table, the desired angular velocity decreases as the drop height increases.

The direction cosine plot and attitude vector trajectory plot at 4.3718 m are analyzed as examples. The reason for choosing this height is that the launch squat angle corresponding to this height is 80 degrees, which is the closest to the experimental launch squat angle commonly applied on the robot. The two plots are analyzed to demonstrate that the rotational speeds obtained in the results meet the expectations of the research. Finally, the change in landing energy absorption after applying IAT at this height was analyzed.

As shown in Figure 11, three resulting curves of the attitude vector's direction cosine versus time are plotted, which represents the evolution of the model's attitude during rotation. The three schematic diagrams in the Figure show that the attitude of the model at the corresponding time attitude ① is the initial state when the model is upside down, at the beginning of the simulation, attitude ② is when the direction vector of the model is parallel to the ground, and attitude ③ is when the model falls to the ground after the flip is completed. At the beginning of the simulation, the direction cosine corresponding to attitude ① is (0, 0, −1), which means that the attitude vector points vertically to the

ground. When the gamma in the direction cosine is 0, the state vector is parallel to the ground, which is attitude ② in the figure. When the gamma in the direction cosine is 1, the attitude vector is upward and perpendicular to the ground, which is attitude ③ in the figure. At touch-down, alpha is −0.0499, beta is 0.0299 and gamma is 0.9983; they are close to 1, 1 and 0, and the error rate for gamma is 0.17%. The landing direction cosine proves that the landing attitude vector is vertically upward, which shows that the model is landing vertically with its lower body.

**Table 3.** Simulated jump height at different launch squat angles.

| Drop Height (m) | IAT Desired Rotational Speed (rad/s) |
| --- | --- |
| 0.0956 | 47.5 |
| 0.1588 | 35 |
| 0.3655 | 25 |
| 0.5986 | 20.2 |
| 0.9207 | 16.2 |
| 1.3109 | 13.5 |
| 1.6858 | 12 |
| 2.0853 | 11 |
| 2.5177 | 10 |
| 2.9327 | 9.3 |
| 3.3263 | 8.6 |
| 3.6641 | 8.5 |
| 3.9694 | 8.3 |
| 4.2078 | 8 |
| 4.3718 | 7.8 |

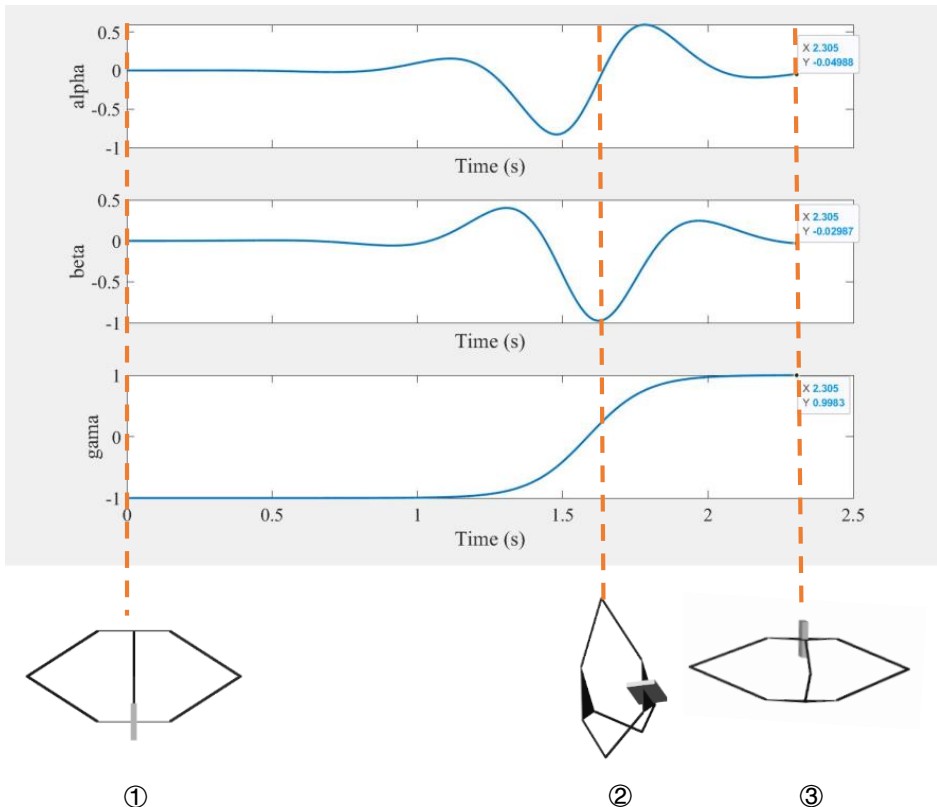

**Figure 11.** Direction cosine plot and three corresponding attitude diagrams of the model.

As shown in Figure 12, the spiral path is the evolution of the attitude vector during falling. The attitude vector at the beginning of the simulation is (−0.0015, 0.0056, −1), approximately (0, 0, −1), which indicates that the robot's attitude is vertically downward.

The attitude vector at the end of the simulation is (−0.0590, 0.0201, 0.9981), approximately (0, 0, 1), which indicates that the robot's attitude is vertically upward. The trajectory spirals up and passes through the origin of the Z axis once, which proves that the model only flips once around the Z axis while rotating.

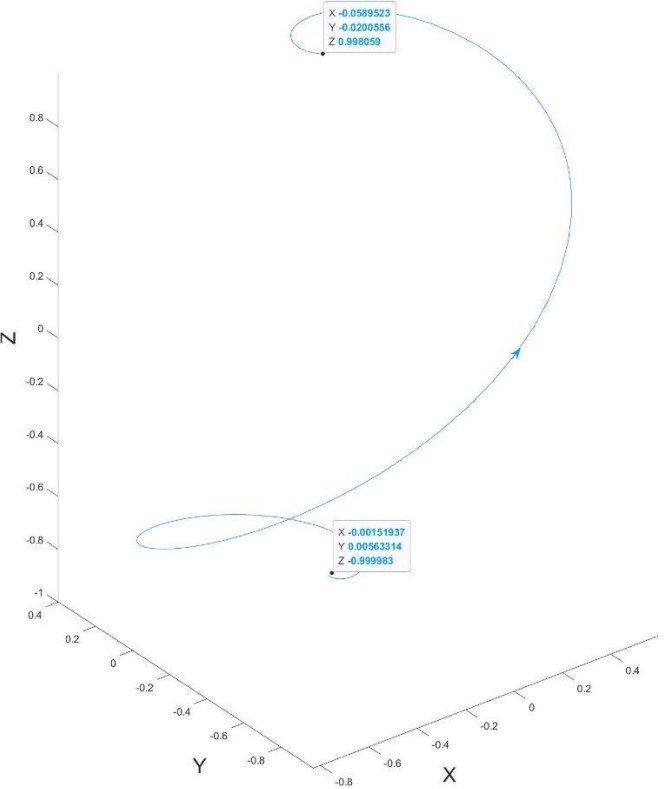

**Figure 12.** Attitude vector trajectory plot.

According to the direction cosine plot and attitude vector trajectory plot, after inputting the desired rotation speed, the robot reversed once during the fall and landed vertically with its lower body. However, there are discrepancies between the data in the two plots and expected values because the input rotational speed of each attempt was accurate to 0.1 rad/s. Since the error is within an acceptable range, it can be concluded that the rotational speeds obtained in the results achieve the expectations of the research.

As shown in the Table 4, the results of the 4.3718 m drop height are calculated through simulation. The results of two different simulation experiments are compared in Table 5. After applying the initial angular velocity, the energy required for a jump increased by 0.0027 joules, the energy absorbed by landing also increased by 0.0482 joules, and the ratio of absorbed energy to energy input increased by 2.76%. Therefore, it can be concluded that the application of the IAT is helpful for both landing attitude control and landing energy absorption.

**Table 4.** Simulation results at 4.3718 m drop height.

| Parameter | Unit | Data |
| --- | --- | --- |
| Desired initial angular velocity | rad/s | 7.8 |
| Rotational kinetic energy | J | 0.0027 |
| Compression energy | J | 1.7294 |
| Total energy input | J | 1.7321 |
| Energy absorbed | J | 0.2792 |
| Energy saving percentage | % | 16.12 |

**Table 5.** Landing energy absorption comparison.

| Parameter | Unit | Free Fall Landing | Laing Applied IAT |
|---|---|---|---|
| Initial angular velocity | rad/s | 0 | 7.8 |
| Total energy input | J | 1.7294 | 1.7321 |
| Energy absorbed | J | 0.2310 | 0.2792 |
| Energy saving percentage | % | 13.36 | 16.12 |

## 6. Conclusions and Future Work

In this paper, by utilizing the robotic simulation system and establishing a reasonable model, the landing dynamics of the simulation model were analyzed. In the simulation, the goal of improving the robot's landing buffer by absorbing landing energy was achieved, and the best landing leg postures under different heights were obtained. In addition, the idea that the application of the IAT could contribute to landing energy absorption and posture control was confirmed in the simulation. The desired initial rotational speeds under different heights were obtained. At the jump height closest to the experimental conditions of the robot, the simulated energy saving ratio was 13.36%, and this ratio increases to 16.12% after applying IAT. Comparing to the other jumping robots, such as "BionicKangaroo" [27], and "KenKen" [9], they have elastic buffering legs similar to three-legged jumping robots to absorb impacts, and their landing energy absorption rates are 20.6% and 18.8%, respectively. The landing energy absorption rates obtained by the simulation experiments in this paper are slightly lower than that of other similar jumping robots, but are still acceptable. In summary, this paper achieved the goal of simulating the landing buffer of the three-legged jumping robot and verifying the methodologies of landing energy absorption and landing posture control. These simulations are helpful to the subsequent optimization, so as to gain a more stable landing buffer to achieve continuous jumps.

Since many results in the simulation experiment were obtained by trial-and-error methodology, the precision of the input values limited the accuracy of the results, resulting in some deviations from the expected values. Although these deviations were within an acceptable range, this error can be reduced, and more accurate results can be obtained by inputting more precise values. To apply the IAT, an initial angular velocity was applied to the model, but the jumping robot is not able to rotate in practice. Therefore, a mechanical system that makes it rotate during launch needs to be developed in the future. In addition, changing the model's knee mass distribution can also cause deviations during rotation. The knee mass distribution can also be explored as a condition for studying the phenomenon of the IAT.

**Author Contributions:** Conceptualization, Y.Y.; methodology, A.M.-R. and Y.Y.; software, A.M.-R. and Y.Y.; validation, K.S.; formal analysis, Y.Y.; investigation, Y.Y.; resources, K.S.; data curation, Y.Y.; writing—original draft preparation, Y.Y.; writing—review and editing, K.S. and H.Z.; supervision, K.S. All authors have read and agreed to the published version of the manuscript.

**Funding:** This research received no external funding.

**Institutional Review Board Statement:** Not applicable.

**Informed Consent Statement:** Not applicable.

**Data Availability Statement:** Not applicable.

**Acknowledgments:** I would like to thank Wenjun (Chris) Zhang, for his professional guidance on this paper. With his critical comments and relevant information provided, I revised my paper to make it more logical and professional.

**Conflicts of Interest:** The authors declare no conflict of interest.

## Appendix A. MATLAB Codes

```matlab
%% Preamble
clear
clc;
close all
addpath(genpath('Libraries'));
%% Simulation
% Physical constants
% Gravitational acceleration
grav_acc = -9.80665*0.166; % Moon kinematic constraints
% Simulation time
sim_time = 2.5; % [s]
% Addition for Intermediate Axis Theorem Experiment
mass_extra = 40.21; % [g]
% Initial angular velocity
omegaz_o = 7.8; % [rad/s]
% Model parameters
% Floor geometry. A slab is used in the simulation as floor. The "slab"
% value sets the edge size of a square. Depth is already defined in the SLX file
slab = 100; % [cm]
% Base plates. One for the upper body of the robot, the other for the foot (lower).
% The "X_base_c" value sets the edge size of an equilateral triangle (three legs).
U_base_c = 6; % [cm]
L_base_c = 6; % [cm]
% Masses. MASS of lower and upper plates. An extra mass accounts for the
% mass of the mechanical power system, i.e., motor, gears, etc.
mass_plate = 8.19; % [g]
% Legs. Three legs made of six rectangular prismatic segments, two each.
% Upper leg segment 1
U_length_L1 = 7.5; % [cm]
U_width_L1 = 0.2; % [cm]
U_thickness_L1 = 0.2; % [cm]
U_mass_L1 = 1.55; % [g]
% Upper leg segment 2
U_length_L2 = U_length_L1; % [cm]
U_width_L2 = U_width_L1; % [cm]
U_thickness_L2 = U_thickness_L1; % [cm]
U_mass_L2 = U_mass_L1;
% Upper leg segment 3
U_length_L3 = U_length_L1; % [cm]
U_width_L3 = U_width_L1; % [cm]
U_thickness_L3 = U_thickness_L1; % [cm]
U_mass_L3 = U_mass_L1;
% Knees. Just a point mass in each leg
U_knee_L1 = 9.39/3; % [g]
U_knee_L2 = 9.39/3; % [g]
U_knee_L3 = 9.39/3; % [g]
% Upper Leg 1
L_length_L1 = U_length_L1; % [cm]
L_width_L1 = U_width_L1; % [cm]
L_thickness_L1 = U_thickness_L1; % [cm]
L_mass_L1 = U_mass_L1;
% Upper Leg 2
L_length_L2 = U_length_L1; % [cm]
```

```
L_width_L2 = U_width_L1; % [cm]
L_thickness_L2 = U_thickness_L1; % [cm]
L_mass_L2 = U_mass_L1;
% Upper Leg 3
L_length_L3 = U_length_L1; % [cm]
L_width_L3 = U_width_L1; % [cm]
L_thickness_L3 = U_thickness_L1; % [cm]
L_mass_L3 = U_mass_L1;
% Stiffness of main driver.
% Note that this element is not shown in the rendered model.
spring_stf = 32.62; % [N/m]
lo = 0; % Natural length [cm]
% Distance to circumcentre of base plates. Refer to the geometrical model
U_base_c = U_base_c/sqrt(3); % [cm]
L_base_c = L_base_c/sqrt(3); % [cm]
% Initial conditions
% Squat angle. This is the angle with respect to the vertical
knee_Angle = 62; % [deg]
% Squat height. For reference points for this see SLX file.
h_0 = 437.18 + 2*(L_length_L1)*cosd(knee_Angle) - (L_thickness_L1/2);
% CALL MODEL AND SIMULATE.
asd = sim('EDRhomboid');
%Spiral path.
ax_v = [u_x.data - l_x.data,u_y.data - l_y.data,u_z.data - l_z.data]; %Direcion vecotr
generated by the center point of upper and lower body
ax_v = ax_v./realsqrt(sum(ax_v.^2,2)); %Switch the direction vector to a unit vector
figure
plot3(ax_v(:,1),ax_v(:,2),ax_v(:,3))
xlabel('X','FontSize',20);
ylabel('Y','FontSize',20);
zlabel('Z','FontSize',20);
axis equal
%Direction cosine
v_x = u_x.data - l_x.data; %X value of direction vector
v_y = u_y.data - l_y.data; %Y value of direction vector
v_z = u_z.data - l_z.data; %Z value of direction vector
abs_v = sqrt(v_x.*v_x + v_y.*v_y + v_z.*v_z); %Distance of the direction vector
alpha = v_x./abs_v; %Diretion cosine on X axis
beta = v_y./abs_v; %Diretion cosine on Y axis
gama = v_z./abs_v; %Diretion cosine on Z axis
f = figure('rend','painters','pos',[10 10 1200 800]);
p = uipanel('Parent',f,'BorderType','none');
p.Title = 'Direction cosine graph';
p.TitlePosition = 'centertop';
p.FontSize = 25;
p.FontWeight = 'bold';
pl(1) = subplot(3,1,1,'Parent',p);
plot(u_x.time,alpha,'LineWidth',2);
axc = get(gca,'XTickLabel');
set(gca,'XTickLabel',axc,'FontName','Times','fontsize',18)
xlabel('Time (s)','FontSize',20);
ylabel('alpha','FontSize',20);
pl(2) = subplot(3,1,2,'Parent',p);
plot(u_x.time,beta,'LineWidth',2);
```

```
axc = get(gca,'XTickLabel');
set(gca,'XTickLabel',axc,'FontName','Times','fontsize',18)
xlabel('Time (s)','FontSize',20);
ylabel('beta','FontSize',20);
pl(3) = subplot(3,1,3,'Parent',p);
plot(u_x.time,gama,'LineWidth',2);
axc = get(gca,'XTickLabel');
set(gca,'XTickLabel',axc,'FontName','Times','fontsize',18)
xlabel('Time (s)','FontSize',20);
ylabel('gama','FontSize',20);
set(pl(1),'xticklabel',[]);
set(pl(2),'xticklabel',[]);
%Rotational kinetic energy
%Moment of inertia of the upper block
I1=1/12*mass_extra/1000*(0.05*0.05+0.005*0.005); %[kg*m^2]
%moment of inertia of the triangle plate
I2=1/12*U_base_c/100*U_base_c/100*mass_plate/1000; %[kg*m^2]
%Moment of inertia of one leg segment
I3=U_mass_L1/1000*(0.5*U_length_L1/100*sind(knee_Angle)+(U_base_c/100)/
sqrt(3))^2+(1/12)*U_mass_L1/1000*(U_length_L1/100*sind(knee_Angle))^2; %[kg*m^2]
%Moment of inertia of one knee point mass
I4=U_knee_L1/1000*(U_length_L1/100*sind(knee_Angle)+(U_base_c/100)/sqrt(3))^2;
%[kg*m^2]
%Total moment of inertia of the model
I=I1+2*I2+6*I3+3*I4; %[kg*m^2]
%Rotational kinetic energy of the model
E=0.5*I*omegaz_o*omegaz_o; %[J]
%%Energy absorption
%Elastic potential energy
%Elastic potential energy of the spring between leg1 and leg2
U12 = 0.5*spring_stf*(-T_12.data/spring_stf).^2; % [J]
%Elastic potential energy of the spring between leg2 and leg3
U23 = 0.5*spring_stf*(-T_23.data/spring_stf).^2; % [J]
%Elastic potential energy of the spring between leg1 and leg3
U13 = 0.5*spring_stf*(-T_13.data/spring_stf).^2; % [J]
%Total elastic potential energy
elastic_potential_energy=U12+U23+U13;
%Energy absorption
elastic_potential_energy_absorb=max(elastic_potential_energy)-min(elastic_potential_
energy);
```

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
