# Peer review of "Simulation of the Landing Buffer of a Three-Legged Jumping Robot"

_machines, doi:10.3390/machines10050299_

Round 1

Reviewer 1 Report

The manuscript deals with an interesting topic "Simulation of the Landing Buffer of a Three-legged Jumping Robot". The work contains a number of grammatical errors. It is also necessary to adjust the word order and grammatical correctness of sentences. The work has the potential for publishing but only after correcting weaknesses. I am choosing a few problematic points that need to be corrected:

  1. Please explain what locomotion stability is
  2. Please unify the reference to the pictures, once you use the abbreviated form Fig. and sometimes debt form Figure.
  3. The manuscript lacks a designed mechanical model of a Three-legged Jumping Robot. Was the robot solved only as a mathematical model and subsequently the experiment consisted only in simulation verification?
  4. It is not necessary to state asterix multiplication in mathematical equations. Leave the multiplication in all equations without the sign of multiplication. In scientific articles, normal multiplication is not indicated if it is not directly related to other operations.
  5.  What geometric and mass characteristics were the inputs in the kinematic analysis?
  6. Please insert Figure 9 in better quality because the legend is very difficult to read.
  7. In Figure 12, the units on the individual axes are missing.
  8. The authors state that there is a positive energy balance when the robot falls to do ground. I recommend testing this simulation model experimentally in the future. 
  9. Is it possible to apply these energy gains to other planets?

Author Response

Thank you for your comments and suggestions. The response to your comments is provided point-by-point below:

  1. Locomotion stability is the ability to move stablely.
  2. The references to the pictures were unified.
  3.  Yes, by far the experiment conducted only in simulation verification.
  4.  The equations have been revised based on your comments.
  5.  The geometric and mass characteristics inputs can be found in the MATLAB codes in Appendix A.
  6.  A better quality figure was inserted.
  7.  Figure 12 shows the change of robot's axial vector. The three axes represent the three directions' unit vector (from-1 to 1), so that's why there is no unit.
  8.  The suggestion of experimentally testing this simulated model is added to future work.
  9.  Yes, its possible to apply the energy absorption to other planets in the simulation by changing the environment characteristic inputs, but the simulation results need to be recalculated.

Reviewer 2 Report

Review of the article

"Simulation of the Landing Buffer of a Three-legged Jumping Robot"

The issues of modelling and simulation of a three-legged jumping robot were present in the paper. The authors simulated the robot's jumping and analyzed the landing buffer of the robot. The paper verifies a method of absorbing landing kinetic energy to improve landing stability and storing it as the energy for the next jump in simulation.

The issues raised in the article are valid. The used computer and testing methods are appropriate, and the results are promising. After rebuilding, the article can be published in the journal after correcting and expanding the content.

  1. A review of the state of knowledge is too short. A complete overview of the knowledge of jumping robots and energy absorption should be done.
  2. The article should clearly define the purpose of the research.
  3. Chapter 2 - the robot's construction is not properly presented. The photos are illegible. The kinematic scheme, principles of operation, mass and geometrical parameters are not given. What was the simulation plan? With what experiment results were the simulation results compared (line 160-162)? 
  4. Figure 6 - What point of the robot is described by the waveforms of height, speed and acceleration? The speed should be negative when the robot is falling. Why is the acceleration constant?
  1. Chapter 4 - energy absorption model was not described, block diagram illegible. How was the force Fx determined? Where in the model is formula (1) used? How was the energy of compression and absorption determined? Quality Fig. 9 is very poor. How were the waveforms in this graph determined?
  2. Chapter 5 - What was the purpose of using IAT in jump simulation? Are Imodel and angle teta constant while jumping? How was the rotation speed w = 7.8 rad/s selected. There is no description of the simulation model taking into account the IAT. Diagram illegible. Where was the formula (8) used? The trajectory of which point on the robot is shown in Fig. 12. How was the directional cosine determined? What do the alpha, beta, gamma angles describe?
  3. Chapter 6 - the conclusions drawn are not in line with the study's objectives. There is no critical discussion of the results achieved in the chapter and no reference to the results of other researchers.

Detailed comments:

  1. The drawings included in the work are of low quality and not legible.
  2. The diagrams provided are poorly described.
  3. Each chart is presented in a different style.
  4. The authors introduce formulas that they do not refer to in work.
  5. There are mistakes in the formulas - e.g. formula (2).
  6. Please correct the text. There are mistakes in the text, e.g. line 43 - "fog" or "frog".

Author Response

Thank you very much for your comments and suggestions, which guided me through the revised manuscript. Point-by-point responses to comments are as follow:

  1. More sufficient background knowledge of jumping robots and energy absorption has been provied in the introduction.
  2. More clearly purpose of the research has been defined at the end of the introduction.
  3. With  more clear pictures added, robot's construction part has been revised to be more detailed. The kinematic scheme, principles of operation, mass and geometrical parameters have been presented more clearly. I don't quite understand what the simulation plan in this point means? Because Chapter 2 mainly introduces the structure and jumping performance of the three-legged robot.A photo of the experimental jumping result of the robot  has been added, and the experimental results that used to compare with the simulation results have been explained more clearly.
  4. Based on your comments, I found the velocity and acceleration waveforms to be meaningless when compared to experimental data. Only the waveform of the height has been kept, and its vertex has been used as the  jumping height of the simulation and compared with the experimental results to verify the reliability of the simulation model. The velocity and acceleration in the original graph were their absolute values, which was why the velocity and acceleration were positive numbers when falling. The acceleration in the graph was not 0 but 9.81.
  5. The energy absorption model was modified on the basis of the simulation model prototype. In the Simulation Procedure, the modifications in the model have been described more detailly, and the specific parameters of the blocks added and modified in Simulink have been displayed. In Simulink, the Sense Force function in the three Spring and Damper Force block was enabled, so that the elastic force Fx on each spring was output. After the data matrix of the elastic force of the three springs over time was output to the WorkSpace, they were added together and (1) was applied to obtain the data matrix of total elastic potential energy changed with time. In the obtained matrix, the begaining total elastic potential energy was considered as the compression energy. Then, by calculating the difference of the elastic potential energy before and after landing, so that the energy absorbed during landing could be obtained. The above calculation process is implemented by the codes in MATLAB, which can be found in the ‘Energy absorption’ paragraph in the appendix. The initial squat angle and initial height of the robot can be set in the MATLAB code of the prototype, which is used to set the landing squat angle and height of the robot. At each drop height, different landing squat angles were input to obtain the corresponding elastic potential energy difference. The landing squat angle was set from 0 degrees to 80 degrees, accurate to 1 degree. Therefore, at each height, the elastic potential energy differences corresponding to the 81 landing squat angles were generated. In this way, the previously obtained 15 drop heights are sequentially substituted into the code so that the relationship between the landing squat angle and energy absorption at different heights was obtained.Finally, the resulting curves for the 15 sets of data were plotted in Figure 9. Figure 9 has been replaced with a clearer picture.
  6. Since the three-legged jumping robot has no aerial righting, the landing attitude is inclined. After landing, it needs to be reset manually to conduct the next jump. The application of IAT is to make the attitude of the robot vertical when landing, so that the previously verified landing buffer method can be applied. Since the robot has been compressed before falling, in order to prevent the robot from decompressing during the fall, the theta angle is limited during the fall, so it remains unchanged. After landing,  robot continues to be compressed due to the energy absorption, the theta angle is increasing at this time. Imodel is changed with angle theta. The calculation of  Imodel is to obtain the energy required for the robot to obtain the desired initial rotation speed. So only the initial Imodel is calculated. When the landing squat angle and drop height remain unchanged, by experimenting to input different rotation speeds until the direction cosines were output 1 for α and β and 0 for γ. The initial rotation speed was tried from 0 rad/s, increasing by 0.1rad/s each time and running the simulation repeatedly. After each simulation, the landing direction cosine of the robot was observed, until the direction cosines were output 1 for α and β and 0 for γ. When the condition of the direction cosine was satisfied, the corresponding vector direction trajectory graph was checked to prove that the model had only flipped once, so that the input angular velocity was proved to be desired. The description of the simulation model for IAT has been modified to be more detailed. After the required angular velocity is obtained, the input energy required to achieve the angular velocity is calculated according to (8). The codes for calculating rotational kinetic energy can be found in the ‘Rotational kinetic energy’ paragraph in the appendix. Figure 12 shows the trajectory of the unit vector formed from the midpoint of the lower body to the midpoint of the upper body. The formula for calculating the direction has been added in the methodology, and the steps for its acquisition have been elaborated in the Simulation Procedure. Alpha, beta, gammaare are the direction cosines for the X axis, Y axis, Z axis(cos a, cos b, cos c).
  7. The discussion has been modified to be more critical. The inadequacies in the research have been summarized, and the results of other researchers have been referenced and compared.

Reviewer 3 Report

The paper "Simulation of the Landing Buffer of a Three-legged Jumping 2 Robot" by Yan et al. describes a modelling and simultion of a three-legged jumping robot".

The article presents crucial issues that need to be addressed.

Also the state of the art is poorley explored.

I suggest to inglude the folollowing references and comment on them in the introduction and in the discussion

Mo, X., Ge, W., Miraglia, M., Inglese, F., Zhao, D., Stefanini, C., & Romano, D. (2020). Jumping locomotion strategies: from animals to bioinspired robots. Applied Sciences10(23), 8607.

Sayyad, A.; Seth, B.; Seshu, P. Single-legged hopping robotics research—A review. Robotica 200725, 587–613.

A deep English revision by a native speaker is needed. 

Author Response

Thank you for your comments and suggestions. I revised the introduction and disscusion based on the two references suggsted by you. The knowledge of jumping robots and energy absorption in introduction is supplemented. In the conclusion, a critical discussion of the results achieved in this paper is added by referring to the results in the two references. A deep English revision will be done by a native speaker.

Round 2

Reviewer 1 Report

With their modifications, the authors eliminated the shortcomings and significantly improved the overall manuscript. After grammar correction, I recommend publishing.

Author Response

Thanks for your comment, the English grammar has been corrected.

Reviewer 2 Report

The corrected version of the article presented for review may be accepted for publication. The changes introduced by the Authors to the text, figures and tables significantly increased the scientific and utilitarian value of the article.
I suggest that the authors consider removing the introduced new figure no. 4 - the figure does not contain any necessary information needed to understand the research.
The article requires a detailed correction of editorial errors - e.g. line 79 - figure caption and chapter title in one paragraph, etc.

Author Response

Thanks for your comments.

The Figure 4 has been removed as your suggestion. 

Since the type of the paper says "All figures and tables should be cited in the main text as Figure 1, Table 1, etc.", I tried to use a different style to avoid this conflict.

Reviewer 3 Report

Authors addressed my comments and now the manuscript is much improved.

However, I think the introduction and the discussion can furrther be refined with previous studies on jumping agents (natural/artificial).

Some suggestion on relevant studies are below

Romano, D., Bloemberg, J., Tannous, M., & Stefanini, C. (2020). Impact of aging and cognitive mechanisms on high-speed motor activation patterns: evidence from an orthoptera-robot interaction. IEEE Transactions on Medical Robotics and Bionics2(2), 292-296.

Scarfogliero, U., Stefanini, C., & Dario, P. (2007, April). Design and development of the long-jumping" grillo" mini robot. In Proceedings 2007 IEEE International conference on robotics and automation (pp. 467-472). IEEE.

Arikawa, K., & Mita, T. (2002, May). Design of multi-DOF jumping robot. In Proceedings 2002 IEEE International Conference on Robotics and Automation (Cat. No. 02CH37292) (Vol. 4, pp. 3992-3997). IEEE.

Please, further check English language.

Author Response

Thanks for your comments, the three pepers have been cited in the manuscript to refine the introduction and the discussion.